# Topical Protease Inhibitor Decreases Anal Carcinogenesis in a Transgenic Mouse Model of HPV Anal Disease

**DOI:** 10.3390/v15041013

**Published:** 2023-04-20

**Authors:** Laura C. Gunder, Hillary R. Johnson, Evan Yao, Tyra H. Moyer, Heather A. Green, Nathan Sherer, Wei Zhang, Evie H. Carchman

**Affiliations:** 1Department of Surgery, School of Medicine and Public Health, University of Wisconsin, 600 Highland Avenue, Madison, WI 53792, USA; gunderl@surgery.wisc.edu (L.C.G.); hjohnson42@wisc.edu (H.R.J.); eyao@wisc.edu (E.Y.); tmoyer2@wisc.edu (T.H.M.); 2Carbone Cancer Center, School of Medicine and Public Health, University of Wisconsin, 600 Highland Avenue, Madison, WI 53705, USA; hagreen@uwcarbone.wisc.edu; 3McArdle Laboratory for Cancer Research and Institute for Molecular Virology, University of Wisconsin, 1111 Highland Avenue, Madison, WI 53706, USA; nsherer@wisc.edu; 4Department of Pathology and Laboratory Medicine, School of Medicine and Public Health, University of Wisconsin, 3170 UW Medical Foundation Centennial Building (MFCB), 1685 Highland Avenue, Madison, WI 53705, USA; wzhang3@kumc.edu; 5William S. Middleton Memorial Veterans Hospital, 2500 Overlook Terrace, Madison, WI 53705, USA

**Keywords:** anal dysplasia, anal cancer, squamous cell carcinoma of the anus, HPV, chemoprevention

## Abstract

Anal cancer is a major health problem. This study seeks to determine if the topical protease inhibitor Saquinavir (SQV), is effective at the prevention of anal cancer in transgenic mice with established anal dysplasia. *K14E6/E7* mice were entered into the study when the majority spontaneously developed high-grade anal dysplasia. To ensure carcinoma development, a subset of the mice was treated with a topical carcinogen: 7,12-Dimethylbenz[a]anthracene (DMBA). Treatment groups included: no treatment, DMBA only, and topical SQV with/without DMBA. After 20 weeks of treatment, anal tissue was harvested and evaluated histologically. SQV was quantified in the blood and anal tissue, and tissue samples underwent analysis for E6, E7, p53, and pRb. There was minimal systemic absorption of SQV in the sera despite high tissue concentrations. There were no differences in tumor-free survival between SQV-treated and respective control groups but there was a lower grade of histological disease in the mice treated with SQV compared to those untreated. Changes in E6 and E7 levels with SQV treatment suggest that SQV may function independently of E6 and E7. Topical SQV decreased histological disease progression in HPV transgenic mice with or without DMBA treatment without local side effects or significant systemic absorption.

## 1. Introduction

High-risk human papillomaviruses (HPVs) cause ~5% of cancers worldwide, with an estimated 630,000 patients diagnosed with an HPV-related cancer annually [1]. While the incidence of HPV-associated cervical cancer (currently ~11,000 cases annually) has declined for more than two decades [1], the incidence of HPV-associated anal cancer in the United States has increased markedly over the same period (now ~7000 cases annually), especially impacting people living with HIV (PLWH). While there is an effective vaccine available to prevent infection by most high-risk HPV strains, vaccination rates in the United States are far from ideal [2] and HPV vaccinations are not therapeutic in those already infected with HPV. Unlike HPV-associated cervical cancer, effective preventative therapies for anal cancer are limited and standard-of-care therapies are associated with painful side effects, as well as high rates of recurrence (>50%) [3].

The currently available treatments for anal dysplasia, the precancerous lesions that progress to anal cancer, include topical treatments and destructive techniques. Topical treatment options, such as 5-fluorouracil (5-FU) or Imiquimod, target cells indiscriminately (HPV+ and HPV−) and induce inflammation in normal and diseased tissues. This results in patient discomfort (skin burning, ulcerations, etc.) and poor compliance. These therapies also have low curative rates, with complete responses of only 17% (5-FU) and 24% (Imiquimod). Furthermore, 27% of 5-FU users and 43% of Imiquimod users experience grade 3–4 toxicities [4]. Additionally, destructive techniques (ablation/excision) for precancerous lesions are invasive and painful, often requiring anesthesia, and can result in long-term anorectal dysfunction such as fecal incontinence and anal stenosis [5]. Due to high recurrence rates of upwards of 50%, costly regular surveillance is required [6]. Presently, there are no national guidelines for screening or treatment of anal dysplasia, primarily due to the lack of effective or well-tolerated therapies. Therefore, the development of a well-tolerated therapeutic for anal cancer prevention is necessary. 

High-risk strains of HPV such as HPV16 and HPV18 cause epithelial cancers including anal cancer through the activities of two viral oncoproteins, E6 and E7, that inactivate the host cellular p53 and pRb tumor suppressor pathways, respectively [7,8,9]. HPV-positive cancers require constitutive expression of E6 and E7; thus, targeted inactivation of these proteins could provide the basis for an anticancer therapeutic strategy [10,11,12,13]. Our group recently demonstrated that a subset of aspartyl protease inhibiting drugs known as protease inhibitors (PIs), which are currently used to treat human immunodeficiency virus (HIV) cause marked reductions in HPV16 E6 and E7 protein levels in two independent cell culture models. These models are HPV16-transformed CaSki cervical cancer cells and NIKS16 organotypic raft cultures, a 3-D biomimetic HPV16-positive model of epithelial pre-cancer. Treatment of CaSki cells with certain PIs (lopinavir, ritonavir, nelfinavir, and saquinavir) reduces E6 and E7 levels and correlates with increases in the expression of the tumor suppressor p53 with a reduction in cell viability [14]. Importantly, these effects also appear to be selective for E6 and E7, with greater toxicity observed for HPV16-positive cells compared to cell lines that are HPV-negative [9].

HIV PIs are known to exhibit generalized anti-cancer properties both in vitro and in vivo, including in the treatment of HPV-associated cancers [15,16,17]. However, the mechanism/s that underpin these effects are still not well understood. Hampson et al., 2016, previously demonstrated strong efficacy of HIV PIs in treating cervical dysplasia in HIV-negative……patients when using a topical application of PIs (lopinavir/ritonavir combination); this demonstrates that direct application of PIs to the cervical epithelium produced an increase in local epithelial concentrations of the drug 10-fold relative to oral dosing [18].

Our in vivo data using mouse papillomavirus further strengthens this premise; we have already observed marked anti-cancer effects for topical SQV with no evidence of tissue damage or systemic effects in mouse models [19]. The rationale for extrapolating from these results and subsequently testing PIs in transgenic mice is that there is a higher rate of cancer development in the transgenic models. Additionally, given that mouse papillomavirus is distinct from HPV it is important to look at this drug in a model that contains the relevant HPV oncoproteins. The tumors that develop in the transgenic mouse models recapitulate the molecular profile of the tumors that we see in humans [20]. We hypothesized that treatment with a topical protease inhibitor, Saquinavir, in transgenic mice would prevent the progression of anal dysplasia to anal cancer while leading to inconsistent E6, E7, p53, and pRb signaling without significant systemic absorption or deleterious side effects.

## 2. Materials and Methods

### 2.1. Mice

For this study, male and female *K14E6/E7* transgenic mice were utilized [21,22,23,24,25,26]. *K14E6/E7* mice constitutively express oncoproteins, E6 and E7, which are associated with a high-risk HPV strain (HPV16), in their epithelium. These transgenic mice develop spontaneous anal dysplasia, in a reproducible way, which can progress to anal cancer. Mice began treatments on Mondays at approximately 25 weeks of age, the age at which 75% or greater of mice develop high-grade anal dysplasia [20]. The mice were monitored weekly during the 20-week treatment period and assessed for the development of overt anal tumors, to determine tumor-free survival. Drug treatments concluded after 20 weeks unless a mouse required sacrifice sooner, per euthanasia criteria. The week following the conclusion of the treatment course, mice were sacrificed, and anal tissue was harvested. Mice that were sacrificed prior to completing 15 weeks of treatment were removed from the study and replaced. 

An approximately equal number of male and female mice were randomized into four treatment groups: control (Male N = 12, Female N = 18, Total N = 30); topical Saquinavir (SQV; Male N = 15, Female N = 25, Total N = 40); topical 7,12-dimethylbenz(*a*)anthracene (DMBA; Male N = 19, Female N = 19, Total N = 38); and topical Saquinavir plus topical DMBA (SQV + DMBA; Male N = 18, Female N = 27, Total N = 45). 

All mice were maintained in the American Association for Accreditation of Laboratory Animal Care-approved Wisconsin Institute for Medical Research (WIMR) Animal Care Facility. The experiments were performed in accordance with approved Institutional Animal Care and Use Committee protocol M006333 and in accordance with the National Institutes of Health guide for the care and use of laboratory animals. 

### 2.2. Topical Saquinavir (SQV)

The dosing concentration of SQV was determined, as in Gunder et al., 2022 [27], using a solution of SQV dissolved in dimethyl sulfoxide and diluted in polyethylene glycol. The lowest concentration of SQV treatment, 2.5%, resulted in a reduction of E6 and E7 oncoprotein expression in the mouse anal tissue (N = 4 mice per SQV dosage group, equal distribution male and female), as assessed by immunohistochemical staining, was selected. Oncoprotein expression was quantified as stated in Section 2.8. See Figure 1. The 2.5% SQV was administered via pipette topically at the anus of each mouse, five days a week for 20 weeks or until mice met the required euthanasia criteria. Control mice were treated with an empty pipette tip to the anus. 

### 2.3. 7,12-Dimethylbenz[a]anthracene (DMBA) Treatment

Topical application of a 0.12 μmol DMBA solution (D3254, Sigma Aldrich, Saint Louis, MO, USA; 60% acetone/40% dimethyl sulfoxide) to the anus of the mice was performed for select treatment groups. Mice were dosed with the DMBA solution once weekly for the 20-week treatment period or until the mice developed overt anal tumors. DMBA was applied at a minimum of 30 min before or after any other topical treatments, to allow for proper absorption of drugs into the anal tissue.

### 2.4. Tumor Assessments

During the weekly thorough visual assessments, initial observations of overt mouse anal tumors were recorded. Mice were gently restrained to obtain measurements of anal tumors (width (W) and length (L)) using calipers. Tumor measurements were recorded weekly until the end of the 20-week treatment or until euthanasia was required.

Tumor volumes (V) were calculated using final tumor measurements prior to sacrifice with the formula [28,29]: V = (W^2^ × L)/2(1)

For mice that had more than one tumor at the anus, tumor volumes were calculated separately, and the sum of the separate tumor volumes was utilized.

### 2.5. Blood and Tissue Collection

Mice were sacrificed following the completion of the topical treatment period. While anesthetized, a minimum of 200 µL of blood was collected from each mouse via cardiac puncture. Mice were sacrificed post-blood draw and anal tissue was harvested and bisected. One section of the anus was fixed in 4% paraformaldehyde solution (as described in Section 2.6) and the other section was frozen fresh at 20 °C for Liquid Chromatography/Mass spectrometry (LC/MS). 

### 2.6. Histology

A portion of anal tissue from each mouse was taken for histology. These portions were fixed in the 4% paraformaldehyde solution for 24 h and then placed in 70% ethanol. After fixation, the tissues were processed, embedded in paraffin, and serially sectioned at 5 μm thickness. Sections were stained with hematoxylin and eosin (H&E) and then evaluated by a trained gastrointestinal pathologist (W.Z.) who was blinded to treatment groups. Sections were evaluated for evidence of anogenital dysplasia or carcinoma. Tissue was scored as normal, low-grade anal intraepithelial neoplasia (LGAIN) (also known as low-grade dysplasia); high-grade anal intraepithelial neoplasia (HGAIN) (also known as high-grade dysplasia); or cancer/squamous cell carcinoma of the anus (anal cancer). Sample embedding, sectioning, and H&E staining were performed by the University of Wisconsin Carbone Cancer Center (UWCCC) Experimental Animal Pathology Laboratory.

### 2.7. Immunohistochemistry (IHC) Staining

IHC staining was performed as previously described by Gunder et al., 2022, using antibodies for human papillomavirus type 16/18 E6 [C1P5] (1:250; GTX20070; GeneTex, Inc., Irvine, CA, USA); human papillomavirus type 16 E7 [6F3] (1:500; GTX60410; GeneTex, Inc., Irvine, CA, USA); p53 [DO-1] (1:350; sc-126; Santa Cruz Biotechnology, Inc., Dallas, TX, USA); and Rb [IF8] (1:50; sc-102; Santa Cruz Biotechnology, Inc., Dallas, TX, USA) [27].

### 2.8. Imaging and Image Analysis

All images were acquired using the Zeiss Axio Imager M2 imaging system at 20× magnification. Images were then analyzed with ImageJ version 2.0.0 (Fiji distribution).

### 2.9. LC/MS Quantitative Analyses

Extractions from mouse sera and anal-tissue samples were analyzed on an Agilent 1100 LC/MSD system and quantified as described in Gunder et al., 2022 [27].

### 2.10. Statistical Analysis

To detect at least a 50% difference in tumor incidence with a type I error rate of 5% and a type II error rate of 20% (80% power) between groups, 12 mice per group were needed. Kaplan–Meier methods and the associated log-rank (Mantel–Cox) tests were run to estimate rates of tumor incidence over time and compare across groups (tumor-free survival). Unpaired t-tests were also performed to assess differences in initial tumor onset and final tumor volumes between DMBA only and SQV + DMBA groups. Chi-square tests or Fisher’s exact tests were used to examine differences between groups (contingency) in the histological grade of tissue samples taken at sacrifice. Ordinary one-way ANOVA with multiple comparisons tests (Šídák’s) was utilized for the analysis of quantitative IHC values. Mean SQV levels in the tissue were identified and compared using unpaired t-tests.

All statistics were performed using GraphPad Prism version 9.4.0 for Mac (GraphPad Software, San Diego, CA, USA). Statistical significance was defined as a *p*-value of 0.05 or less. For all testing, the overall type I error rate was controlled by the omnibus test for an association between the factor and the outcome of interest. Additional pairwise comparisons were unadjusted.

## 3. Results

### 3.1. Gradient Curve

Refer to Figure 1 for the IHC gradient curve of SQV treatment for E6 and E7 expression. Based on the results of Figure 1, 2.5% SQV was selected for topical dosing of all treatment groups as it was the lowest dose that influenced E6 and E7 expression. This trend was not significantly different as only four mice were used per treatment group.

### 3.2. Side Effect Profile and Drug Concentrations

Mice were observed five days per week during dosing and thoroughly evaluated once a week for tumor size measurements. No local side effects were noted with topical SQV treatment, such as hair loss, erythema, or irritation. No systemic side effects with SQV were noted either, such as weight loss, lethargy, or changes in social behaviors (grooming, etc.). The lack of local or systemic side effects was consistent at all doses in the gradient curve.

To evaluate the proper absorption of topical SQV into the desired tissue, anal tissue was harvested and processed. SQV levels were quantified via electrospray ionization mass spectroscopy. The mean tissue concentration of SQV in the treated mice, with and without DMBA, was 236.5 mg/g. This concentration was slightly higher in female mice compared to males (*p* = 0.7715). No drug was detected in the serum in most mice (34/35). A single male mouse in the SQV-only group had a serum SQV level above the lower limit of quantification (LLOQ, 5 ng/mL), at 50.7 ng/mL of SQV. See Table 1. 

### 3.3. Mice Treated with SQV Did Not Demonstrate Improved Tumor-Free Survival 

To assess whether SQV effectively increased tumor-free survival, mouse anuses were inspected thoroughly for the development of overt anal tumors. Two control mice developed overt anal tumors without the utilization of DMBA (2/30) while none of the mice treated with SQV only (0/40) developed overt anal tumors (*p* = 0.1801). Approximately 89.5% of the DMBA-only treated mice (34/38) developed overt anal tumors as compared to 71.1% of mice treated with SQV + DMBA (32/45) (*p* = 0.0556). Of the mice that displayed visible anal tumors, the average onset for DMBA-only treated mice was 16.3 weeks as compared to 15.5 weeks in SQV + DMBA treated mice (*p* = 0.2824). When comparing tumor-free survival curves, there were no significant differences between SQV-only and control groups (*p* = 0.1000) or SQV + DMBA and DMBA-only groups (*p* = 0.5792). See Figure 2. Male and female mice were evaluated separately and there was no statistically significant difference between male and female tumor-free survival in the treatment groups.

### 3.4. SQV Led to Regression of Histologic Disease in Female Mice

To assess the effect of topical SQV treatment on histologic disease, anal tissue was harvested from mice and evaluated for the final histologic grade of disease: normal, low-grade anal dysplasia, high-grade anal dysplasia, or anal cancer. Approximately 76.7% of control mice (23/30) ended the study with high-grade anal dysplasia as compared to 42.5% (17/40) of the mice treated with SQV only, which is expected from our previous work [22]. There was a statistically significant lower grade of disease in the SQV-only mice compared to control mice (*p* = 0.0093). When mice were separated by sex, this outcome remained significant in females (*p* = 0.0003), but not in males (*p* = 0.5159). A single male mouse treated with SQV only (1/40) developed microscopic anal cancer. 

Most mice treated with DMBA alone (81.6% or 31/38) developed anal cancer while less than half (46.7% or 21/45) of mice treated with DMBA in combination with SQV had anal cancer at the end of the treatment period. The remaining 18.4% of DMBA-only treated mice (7/38) exhibited high-grade dysplasia while 42.2% (19/45) and 11.1% (5/45) of the SQV + DMBA group exhibited high-grade and low-grade anal dysplasia respectively. Thus, there was a statistically significant lower grade of disease in the SQV + DMBA mice tissue compared to the DMBA-only mice (*p* = 0.0025). Again, when separating mice by sex, this effect was only significant in female mice as compared to male mice (*p* = 0.0133 and *p* = 0.2345, respectively). See Figure 3.

### 3.5. SQV Did Not Mediate Significant Changes in Overall Oncoprotein Expression or Their Targets

Viral oncoproteins E6 and E7, required for HPV-associated oncogenesis, were evaluated by immunohistochemical staining of anal tissue. There were no statistically significant differences in viral expression for E6 or E7 between treatment groups and respective control groups (E6—control to SQV only *p* = 0.1333, DMBA only to SQV + DMBA *p* = 0.7825; E7—control to SQV only *p* = 0.2428, DMBA only to SQV + DMBA *p* = 0.7891). When separated by sexes, only E6 expression in male tissue was significantly lower in the control group as compared to SQV-only group (*p* = 0.0121). See Figure 4.

Downstream effector proteins of E6 and E7, p53 and pRb, respectively, were examined. There was no significant change in expression in p53 or pRb between control and SQV-only treated mice. However, tissue from mice treated with SQV + DMBA compared to DMBA alone showed a statistically significant decrease in p53 and pRb (p53 *p* = 0.0118 and pRb *p* = 0.0179). When separating by sex, this effect only remained statistically significant in tissue from male mice (p53 *p* = 0.0026 and pRb *p* = 0.0359). See Figure 5.

## 4. Discussion

Preventative therapy for anal cancer, beyond HPV vaccination, is essential given the increasing rates of disease, despite the wide availability of the vaccine [30]. The vaccine is not therapeutic for patients already infected with HPV. Furthermore, our current preventative therapies for anal cancer, by treatment of anal dysplasia, are ineffective and poorly tolerated [5]. The lack of efficacy may be related to the non-molecularly targeted nature of our current anal dysplasia treatments for HPV-infected tissues. Given the previous literature demonstrating the targeted effects of protease inhibitors on HPV oncoproteins, we aimed to examine the role of topical protease inhibitors in a transgenic mouse model of anal disease [14,15,16,31,32,33,34]. 

It is already known that topical delivery of protease inhibitors (lopinavir and ritonavir) to treat HPV-associated precancerous lesions is feasible. A clinical trial looking at HPV-associated dysplasia of the cervix demonstrated that protease inhibitors are effective in causing regression of cervical intraepithelial neoplasia (cervical dysplasia) [18]. This important finding provides a strong rationale for our research. This previous clinical trial also indicated that the pharmacological principles of protease inhibitors make them amenable to topical therapy. First, our focus of localized therapy (topical treatment to the anus) was to decrease the odds of systemic side effects associated with therapy, especially since anal dysplasia is a localized disease. Second, systemic protease inhibitors are linked to moderate patient-associated toxicities (lipid dysfunction, diarrhea, etc.); for this reason, most treatment plans for HIV therapy no longer include protease inhibitors. Finally, retrospective studies have identified systemic protease inhibitors as a potential risk factor for anal cancer development [34,35,36]. The basis for these findings is unclear, but it is critical to note that patients in these studies were taking systemic protease inhibitors for prolonged periods. Importantly, we have shown that topical application of the protease inhibitor, Saquinavir, did not generally appear in the mouse sera collected prior to sacrifice. These data suggest that topical treatment did not lead to systemic absorption of the study drug. A sample from a single mouse had quantifiable levels of SQV in the serum. Possible reasons for this may be due to tool/collection contamination or could be explained by mouse behavior. Mice grooming, including the ability to lick their own anus and those of their cage mates, may have led to oral ingestion of SQV. 

Treatment with topical SQV showed significant improvements in disease in transgenic *K14E6/E7* female mice in terms of regression of histologic disease in mice treated with SQV alone, and a decrease in cancer development in mice treated with SQV in the setting of DMBA treatment. As expected, most of the control *K14E6/E7* mice developed high-grade anal dysplasia [23,24,25,26]. Excitingly, the majority of mice treated with topical SQV alone displayed normal or low-grade dysplasia final anal histology. When separating mice by sex, this outcome only remained significant in female mice, highlighting a clear sex-specific effect. Again, as expected, the majority of mice treated with DMBA alone developed microscopic anal cancer. Importantly, less than half of mice treated with DMBA in combination with SQV progressed to anal cancer. Consistent with non-DMBA treated mice, this outcome was only significant in female mice when separated by sex. The reason for these observed sex-related differences will be an area of future investigation. Future studies will examine the relevance of varying treatment dosages based on total mouse body weight, given female mice tend to be smaller than male mice, and sex-specific hormones such as evaluating the role of estrogen on therapeutic response.

The absence of changes in E6 and E7 oncoprotein expression noted in this work suggests an alternative molecular trajectory, given that SQV treatment ameliorated disease in a population of the treated mice. The lack of change in E6 and E7 expression was corroborated by p53 and pRb expression in this study, the downstream targets of E6 and E7. Our findings were inconsistent with the IHC performed on tissue from the gradient curve mice in which a decrease in E7 expression was observed with topical 2.5% SQV treatment. The discrepancy may be due to the variation in treatment timing as mice in the gradient curve were treated for only two weeks as opposed to 20 weeks. A time course study will be designed to elucidate the possible changes in E6 and E7 expression at varying time points throughout a topical protease inhibitor treatment course. 

Another potential reason that treatment with Saquinavir did not effectively reduce E6 and E7 expression in this study may be related to the lack of immune response to “self” proteins in these mice. This *K14E6/E7* mouse model has constitutive activation of E6 and E7 oncogenes in mice from birth. Thus, E6 and E7 are not recognized as “foreign” and do not elicit an immune response to promote viral clearance as would be seen in humans [20,21]. To further elucidate the importance of the immune system in the treatment response, we plan to utilize conditional HPV16 E6 and E7 mice, which allow for temporal regulation of E6 and E7 activation [37]. These transgenic mice, when exposed to Cre recombinase, will express E6 and E7 at the location of Cre infection. With this model, we will be able to identify the role of immune system stimulation and the interplay between topical protease inhibitors. Additionally, SQV treatment may have had a minimal effect on E6 and E7 protein expression in this specific model due to the type of E6/E7 transcriptional regulation utilized in this mouse model. Here, expression is based on a K14 promoter. If the effect of SQV was on the HPV locus control region (LCR) promoter, the coding region prior to E6 and E7, and not the K14 promoter, there may be a discernible difference in E6/E7 expression.

Overall, we found that topical SQV is well-tolerated, results in robust local tissue concentrations is not consistently systemically absorbed, and is effective in preventing disease progression, even allowing for disease regression without affecting E6 and E7. These conclusions indicate that the protease inhibitor, Saquinavir, likely modulates additional molecular pathways to prevent cancer. We are currently performing proteomic analysis of PI-treated versus non-PI-treated tissue that is HPV positive versus HPV negative to explore other potential pathways of treatment effect.

## 5. Conclusions

Topical SQV decreases histological progression of anal disease in HPV transgenic mice, especially in female mice, regardless of DMBA (topical carcinogen) treatment without leading to local side effects or consistent systemic absorption.

## Figures and Tables

**Figure 1 viruses-15-01013-f001:**
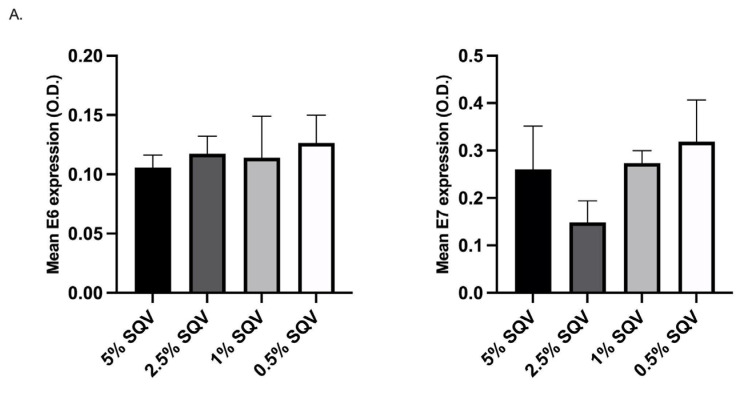
E6/E7 oncoprotein expression (optical density (O.D.)) in mice treated with SQV at doses, 5%, 2.5%, 1%, and 0.5% to establish a standard dosing concentration. (**A**) Mean E6 and E7 expression quantified from N = 4 mice per SQV dosage group, each distribution male and female; (**B**) Representative images of anal tissue from the assigned dosages immunohistochemically stained for E6 and E7.

**Figure 2 viruses-15-01013-f002:**
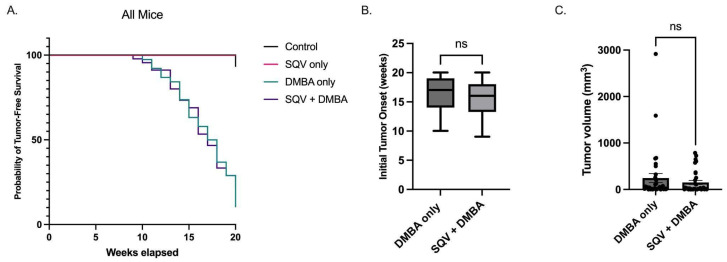
Tumor-free survival, tumor volume, and tumor onset: (**A**) Tumor-free survival and tumor volume over the 20-week treatment period; (**B**) Average initial tumor onset (in weeks) for mice that developed overt anal tumors during the treatment period; (**C**) Final tumor volume measured prior to sacrifice for mice that developed tumors within 20 weeks. In (**B**,**C**), “ns” is not significant.

**Figure 3 viruses-15-01013-f003:**
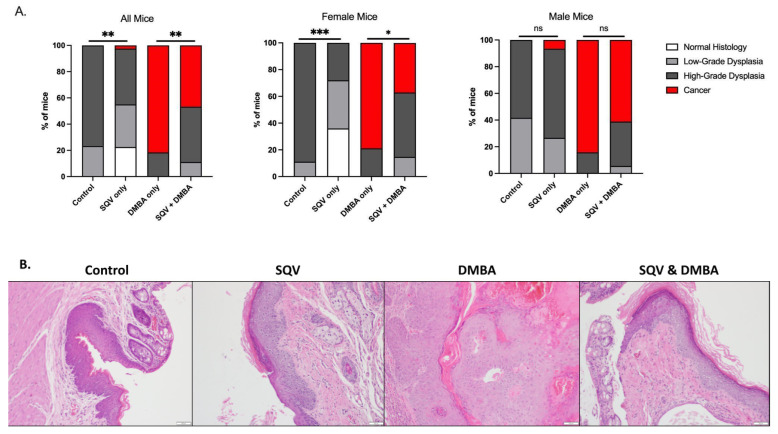
Final anal-tissue histology for mice in each treatment group: (**A**) All mice anal histology (combined female and male mice), female mice only anal histology, and male mice only anal histology. (**B**) Representative histological images per treatment group. Significance was assessed as * *p* < 0.05, ** *p* < 0.01, *** *p* < 0.001, “ns” is not significant.

**Figure 4 viruses-15-01013-f004:**
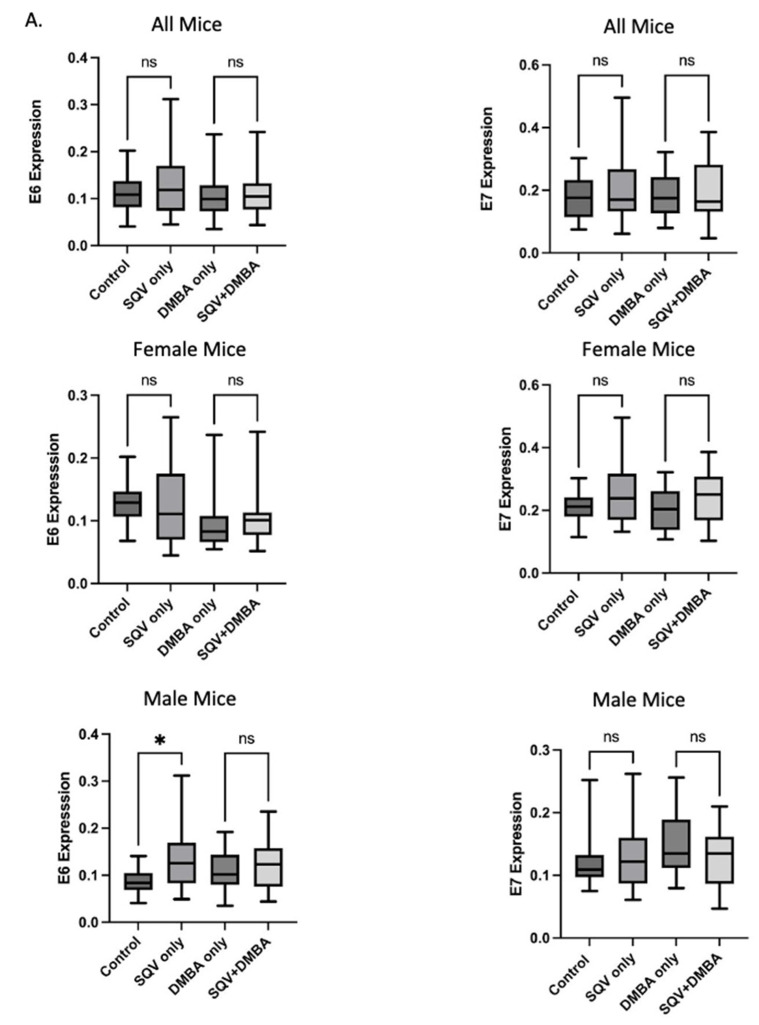
Expression of viral E6 and E7 oncoproteins: (**A**) Quantification of E6 and E7 expression in each treatment group. Significance was assessed as * *p* < 0.05, “ns” is not significant. (**B**) Representative images of immunohistochemically stained anal tissue from mice in each treatment group for E6 and E7.

**Figure 5 viruses-15-01013-f005:**
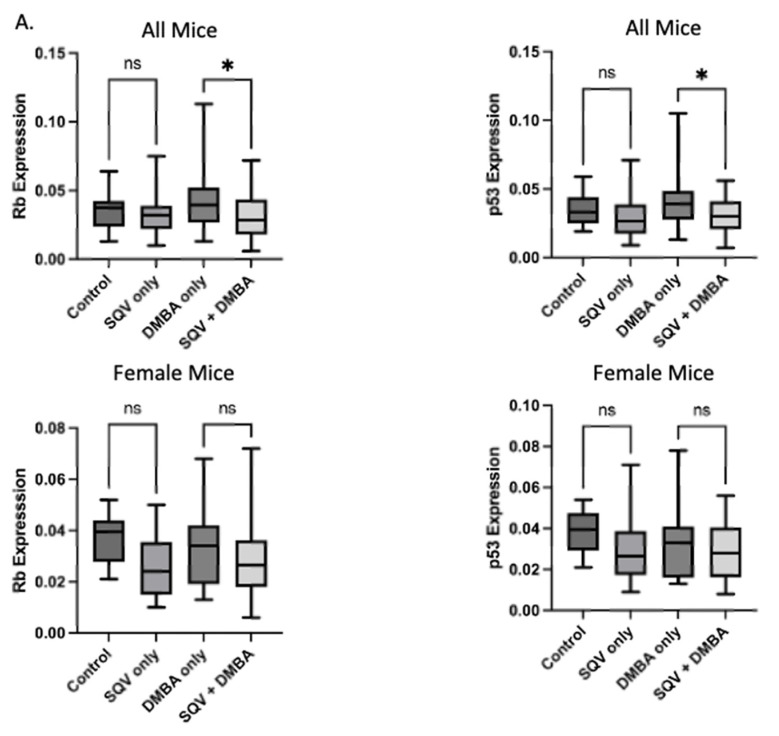
Expression of Rb and p53: (**A**) Quantification of Rb and p53 expression in each treatment group and separated by sex; (**B**) Representative images of immunohistochemically stained anal tissue from mice in each treatment group for Rb and p53. Significance was assessed as * *p* < 0.05, ** *p* < 0.01, and “ns” is not significant.

**Table 1 viruses-15-01013-t001:** Quantification of SQV present in harvested anal tissue and sera from female and male mice.

		mg SQV Per g of Tissue			ng SQV Per mL of Sera		
Treatment Group	*N*	Mean	SD	SEM	Mean	SD	SEM
SQV-treated mice	35	236.5	213.3	36.1	ND *	NA *	NA *
SQV-treated mice (Female)	20	245.7	214.8	48.0	ND	NA	NA
SQV-treated mice (Male)	15	224.1	218.22	56.3	ND *	NA *	NA *

ND = not detectable, * a single SQV only male mouse > 5 ng/mL (LLOQ): 50.7.

## Data Availability

Not applicable.

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
