# Peer review of "Topical Protease Inhibitor Decreases Anal Carcinogenesis in a Transgenic Mouse Model of HPV Anal Disease"

_viruses, 2023, doi:10.3390/v15041013_

Round 1

Reviewer 1 Report

Gunder et al have conducted a study to evaluate the protective effects of saquinavir against anal dysplasia and tumorogenesis in a transgenic mouse model. The manuscript is well written and easy to follow. While their findings indicate that topical saquinavir treatment decreased, or rather delayed histological disease progression, there was no differences in tumor-free survival between the saquinavir-treated and the control group. I have some comments that I hope the authors can address:

- In line 133: "Control mice were treated with an empty pipette tip to the anus". As saquinavir is dissolved in DMSO, it would have been more accurate to treat the control group with equal amounts of DMSO. It is known that DMSO penetrates cell membrane, and can affect cellular viability, inflammatory response and even the extracellular collagen. How can the authors rule out if the DMSO was implicated in any of the observed findings in the saquinavir group?

- In line 77: "Treatment of CaSki cells with certain PIs (Lopinavir,

78 Ritonavir, nelfinavir, and Saquinavir) reduces E6 and E7 levels and correlates with increases in the expression of the tumor suppressor p53 and a reduction in cell viability"

While the authors provide ample explanation of the possible factors that may explain the discrepancy between their findings and other's findings, it would have been important to assess the cell viability using the concentration of the saquinavir used for the experiments in molar concentration. Saquinavir is one of the most toxic of the protease inhibitors at high concentrations on the cellular level, therefore, I hope the authors can provide and insight on the molar concentration used. 

- What is the denominator on the Y axis in figure 1?

- In line 345: "in which a decrease in E6 and E7 expression was observed", from figure 1, given the large deviations, it appears that there was no significant decrease in E6 levels resulting from treatment with saquinavir, did the authors find a statistically significant difference?

- The styling of the references is not uniform

Author Response

  1. In line 133: "Control mice were treated with an empty pipette tip to the anus". As saquinavir is dissolved in DMSO, it would have been more accurate to treat the control group with equal amounts of DMSO. It is known that DMSO penetrates cell membrane, and can affect cellular viability, inflammatory response and even the extracellular collagen. How can the authors rule out if the DMSO was implicated in any of the observed findings in the saquinavir group? 

Thank you for this comment. We have done studies with DMSO alone in mice as the control and given the irritative effects of DMSO the mice did not tolerate it well. Rademacher et al, 2018; PMID 30123096 . We have data from those mice, not published, demonstrating that they do not affect the pathways that we examined in this manuscript. Given the irritative effects and lack of effect on E6, E7, mTOR and PI3K, we did not use DMSO as a control in this experiment.

  1. In line 77: "Treatment of CaSki cells with certain PIs (Lopinavir, 

78 Ritonavir, nelfinavir, and Saquinavir) reduces E6 and E7 levels and correlates with increases in the expression of the tumor suppressor p53 and a reduction in cell viability"

While the authors provide ample explanation of the possible factors that may explain the discrepancy between their findings and other's findings, it would have been important to assess the cell viability using the concentration of the saquinavir used for the experiments in molar concentration. Saquinavir is one of the most toxic of the protease inhibitors at high concentrations on the cellular level, therefore, I hope the authors can provide and insight on the molar concentration used. 

Very interesting points. From our previous paper (Park et al 2021 PMID 33668328), in NIKS RAFT culture we did not see any toxicity at 5 or 10 μM of saquinavir. Given that we used 2.5% Saquinavir for these experiments that equates to 0.37x 10-5 μM, which is much lower than the amount used in RAFTS. We also saw no evidence of toxicity with Saquinavir in terms of redness, blistering, hair loss, or irritation from this specific dose.

  1. What is the denominator on the Y axis in figure 1?

  The denominator, O.D., is Optical Density. The figure axis has been updated appropriately.

  1. In line 345: "in which a decrease in E6 and E7 expression was observed", from figure 1, given the large deviations, it appears that there was no significant decrease in E6 levels resulting from treatment with saquinavir, did the authors find a statistically significant difference?

Thank you for this comment. Only trends in reduction were noted. There were no significant decreases in E6 or E7 levels between groups as assessed by one-way ordinary ANOVAs with Tukey’s multiple comparisons. Given the small number of mice per group, 4, none of the changes were statistically significant. This was clarified in section 3.1., “This trend was not significantly different as only four mice were used per treatment group.”

  1. The styling of the references is not uniform.

Thank you for noting this, the references have been updated to make the style uniform.

Reviewer 2 Report

Gunder et al continue to study the impact of repurposing a protease inhibitor (SQV) used to treat HIV for HPV dependent cancers. This time, they focus on the impact of topical application using an anal cancer model with K14 E6/E7 transgenic mice.

The study looks reasonably straightforward and the manuscript is in general well written.

The major conclusions are that the drug appears well tolerated locally, does not appear to significantly enter the serum and reduces the histological grade of the DMBA induced tumors.

Minor Suggestions:

1) Figure 1.  Must add statistical analysis.  Is anything here actually significant? What does the axis label value represent? Is this arbitrary units or relative to untreated controls? From the experimental description, there no indication that a no SQV control was done for this experiment, so I assumed that the axis is arbitrary units.  I can't see how there is any possibility that SQV reduced E6 expression for any of the dosing levels compared to the lowest concentration (in contrast to what is stated on line 345). If it is relative to a no drug control treatment, the interpretation is completely different. Added to this, no effect on E6 or E7 expression is seen in figure 4, which includes a no SQV control.

2) The y-axis label for figure 2A (Survival) is misleading to this reviewer.  This is not a survival endpoint (ie the mice die).  I think that this actually represents time to first evidence of disease. Not sure what mice researchers do for this, but in human trials, this would perhaps be Disease Free Interval. I suggest changing this axis label to something more informative.

3) From what I can tell, previous work with SQV showing an impact on E6/E7 expression has used normal E6/E7 transcriptional regulation from the HPV LCR.  This is different from the K14 regulation in the transgenic mouse model.  If the effect of SQV was via effects on LCR promoter regulation, I would not be surprised that a reduction of E6/E7 was not observed with the K14 promoter.  This might be added to the discussion, especially as I don't think the statistics will support a significant reduction of E6/E7 in figure 1 as described earlier in my review.   

Author Response

1) Figure 1.  Must add statistical analysis.  Is anything here actually significant? What does the axis label value represent? Is this arbitrary units or relative to untreated controls? From the experimental description, there no indication that a no SQV control was done for this experiment, so I assumed that the axis is arbitrary units.  I can't see how there is any possibility that SQV reduced E6 expression for any of the dosing levels compared to the lowest concentration (in contrast to what is stated on line 345). If it is relative to a no drug control treatment, the interpretation is completely different. Added to this, no effect on E6 or E7 expression is seen in figure 4, which includes a no SQV control.

Thank you for your comments. There were only 4 mice per group in Figure 1, thus there were no statistically significant differences. You are correct, these are arbitrary units and not relative to no treatment controls. We have corrected line 345. Thank you for noticing our oversight. 

2) The y-axis label for figure 2A (Survival) is misleading to this reviewer.  This is not a survival endpoint (ie the mice die).  I think that this actually represents time to first evidence of disease. Not sure what mice researchers do for this, but in human trials, this would perhaps be Disease Free Interval. I suggest changing this axis label to something more informative.

Thank you for this suggestion. The y-axis on the Figure 2A graph has been changed to more accurately represent Tumor-Free Survival.

3) From what I can tell,  This might be added to the discussion, especially as I don't think the statistics will support a significant reduction of E6/E7 in figure 1 as described earlier in my review.

Thank you for your comment, this has been added in the discussion.